# Estimation and implications of the genetic architecture of fasting and non-fasting blood glucose

Zhen Qiao[1,2], Julia Sidorenko [2], Joana A. Revez [2], Angli Xue[1,2], Xueling Lu[3,4], Katri Pärna[3,5], Harold Snieder [3], Lifelines Cohort Study*, Peter M. Visscher [2], Naomi R. Wray [2,6] & Loic Yengo [2] ✉

The genetic regulation of post-prandial glucose levels is poorly understood. Here, we characterise the genetic architecture of blood glucose variably measured within 0 and 24 h of fasting in 368,000 European ancestry participants of the UK Biobank. We found a near-linear increase in the heritability of non-fasting glucose levels over time, which plateaus to its fasting state value after 5 h post meal ($h^2$ = 11%; standard error: 1%). The genetic correlation between different fasting times is > 0.77, suggesting that the genetic control of glucose is largely constant across fasting durations. Accounting for heritability differences between fasting times leads to a ~16% improvement in the discovery of genetic variants associated with glucose. Newly detected variants improve the prediction of fasting glucose and type 2 diabetes in independent samples. Finally, we meta-analysed summary statistics from genome-wide association studies of random and fasting glucose ($N$ = 518,615) and identified 156 independent SNPs explaining 3% of fasting glucose variance. Altogether, our study demonstrates the utility of random glucose measures to improve the discovery of genetic variants associated with glucose homeostasis, even in fasting conditions.

Type 2 diabetes (T2D) is a complex disease characterized by sustained elevation of blood glucose levels, primarily caused by insulin resistance and beta-cell dysfunctions[1,2]. Over the last decades, large-scale association studies have shown that both genetic and environmental factors contribute to glucose homeostasis and T2D pathogenesis[3–7]. In particular, genome-wide association studies (GWAS) of glycemic traits have provided insights into the genetic regulation of glucose levels and that of T2D susceptibility[8–10], while revealing a partial overlap between them[3,11]. Overall, GWAS of glycemic traits have discovered a range of genetic loci associated with fasting glucose (FG) concentration[9,10,12–15], post-challenge glucose concentration[8] and fasting insulin concentration[10,13,15].

Previous theoretical and empirical studies have established that sample size is the main factor driving GWAS discoveries[16,17]. However, oral glucose tolerance tests or tests performed in a fasting state (i.e., fasting for at least 8 h) can be difficult to schedule in practice, which may limit the sample size attainable. Therefore, it is critical to leverage data that are more conveniently collected such as random glucose measures (RG: i.e., blood glucose levels measured at any time of the day, irrespective of the fasting duration), readily available in large population-based cohorts like the UK Biobank (UKB)[18].

In this study, we conduct genetic analyses on RG measured in individuals without diabetes from the UKB to provide an overview of its genetic architecture and evaluate its applicability to improve our

[1]Garvan Institute of Medical Research, Darlinghurst, NSW, Australia. [2]Institute for Molecular Bioscience, The University of Queensland, Brisbane, Australia. [3]Department of Epidemiology, University of Groningen, University Medical Center Groningen, Groningen, Netherlands. [4]Laboratory of Environmental Medicine and Developmental Toxicology, Shantou University Medical College, Guangdong, China. [5]Institute of Genomics, University of Tartu, Tartu, Estonia. [6]Queensland Brain Institute, The University of Queensland, Brisbane, Australia. *A list of authors and their affiliations appears at the end of the paper. ✉e-mail: l.yengodimbou@uq.edu.au

understanding of T2D pathogenesis as well as our ability to predict T2D risk. Our aims are threefold: (1) quantify the heterogeneity in genetic effects on the regulation of blood glucose levels across different fasting time; (2) test how to better model the effect of the duration of fasting status when estimating the genetic effects on RG levels, and (3) examine the utility of RG in the prediction of fasting glucose levels and T2D risk.

## Results

### Overview of study design and phenotype definition

A flow chart illustrating the overview of the study design is provided in the Supplementary material (Supplementary Fig. 1). We used untransformed glucose levels measured in serum samples from UKB participants. Following previous genetic studies of glycemic traits from the Meta-Analysis of Glucose and Insulin-related traits Consortium (MAGIC), we restricted our analyses to nondiabetic samples of European descent (Methods, Supplementary Data 1). Each participant had also reported the time since their last consumption of food or drink intake, hereafter referred to as fasting time. Individuals who had a fasting time greater than 24 h (less than 0.01% of the sample) were defined as outliers and excluded from the analysis. In total, we retained 367,427 individuals for our main analyses, of whom 280,962 were unrelated (i.e., SNP-based estimated relatedness < 0.05; Supplementary Methods, Supplementary Data 2). The mean random glucose level was 4.96 mmol/L with a standard deviation (SD) of 0.63 mmol/L.

### Time-dependent genetic architecture of RG and optimal GWAS strategy

To facilitate the investigation of the genetic architecture of RG within different fasting time intervals, we subset UKB participants into five groups based-on their self-reported fasting time (0–2 h, 3 h, 4 h, 5 h and 6–24 h). These five groups were defined to include at least 30,000 individuals each (Supplementary Data 2, Supplementary Fig. 2). We estimated the SNP-based heritability of each measure using the Haseman-Elston (HE) regression[19] method (Methods). Heritability estimates were positively correlated with fasting time and ranged from 0.05 (standard error; s.e. 0.005) to 0.11 (s.e. 0.01) (Fig. 1a, Supplementary Data 3). The smallest SNP-based heritability estimate was observed in the 0–2 h group, which implies a higher relative contribution of non-genetic effects, e.g., food content and intake.

Next, we estimated the pairwise genetic correlations ($r_g$) between subgroups using linkage disequilibrium (LD) score regression (LDSC)[20] and HE regression and found that the genetic correlation between subgroups is not significantly different from 1 ($P > 0.05$, Fig. 1b, c) for most comparisons except between 0-2 h and 5 h subgroups (Table 1 and Fig. 1c). In addition, we collected GWAS summary statistics of FG from Lagou et al. (2021) ($N = 151,188$ European ancestry individuals without diabetes)[12], and estimated $r_g$ between FG and glucose levels in each subgroup using LDSC. Similarly, we found that $r_g$ estimates between RG and FG are either close to 1 or not significantly different from 1 (Fig. 1d, Supplementary Data 4). Altogether, we conclude from these analyses that the genetic regulation of glucose levels is largely constant across durations of fasting status.

This conclusion suggests that more statistical power in GWAS analyses can, in principle, be achieved from jointly analysing glucose levels at all time points. Therefore, we assessed two simple analytical approaches to achieve this goal. The first one is an inverse-variance weighted meta-analysis of estimated SNP effects across time points (hereafter referred to as the meta-GWAS approach), and the second one is a direct estimation of SNP effects in the entire sample, while correcting for fasting time as a categorical covariate (hereafter referred to as the mega-GWAS approach).

We performed both mega- and meta-GWAS in the UKB and compared the statistical power between these two approaches using the mean chi-square association statistic as well as the number of independent associations detected ($P < 5 \times 10^{-8}$) using conditional and joint analysis (COJO) analysis[21]. The mean chi-square association statistic was 1.46 in the mega-GWAS vs. 1.53 in the meta-GWAS. We also compared the LDSC intercept (a statistic reflecting the degree of confounding in a GWAS; Methods) between these analyses and found that both analyses yielded similar estimates of the LDSC intercept (Table 2, Supplementary Data 5), suggesting that the increased chi-square statistic observed in our meta-GWAS reflects enhanced statistical power but no inflation of false positives. Consistently, we identified 109 and 127 independent associations (COJO SNPs, 72 in common) with glucose levels using our mega-GWAS and meta-GWAS respectively (Table 2, Supplementary Data 6), which represents a 16% increase in the number of signals detected.

We found genome-wide significant evidence of heterogeneous SNP effects across fasting time groups for rs1881415 ($P_{HET} < 5 \times 10^{-8}$; Supplementary Data 7). This variant is in high LD ($r^2 > 0.8$) with SNPs previously associated with fasting glucose, fasting proinsulin, 2h-glucose[22] and T2D risk[23]. Importantly, the T2D risk allele at this locus shows opposite effects on FG (positive effect) vs 2 h glucose levels (negative effect). Our data recapitulates this heterogeneous pattern (i.e., effect sizes are not sign-consistent across fasting times) although the relatively small number of UKB participants fasting for > 5 h reduces the statistical power to detect a genome-wide significant effect.

In summary, we provide empirical evidence that a fasting-time stratified meta-GWAS is optimal for discovery of glucose-associated genetic variants, and therefore focus on this approach in following sections.

### Residual differences in genetic control of FG and RG

We used LDSC to estimate the genetic correlation between glucose (FG and RG) and 245 traits with GWAS summary statistics available in LD hub[24] (Supplementary Data 8). Overall, these 245 traits showed similar genetic correlations with FG and RG (Fig. 2). After Bonferroni correction accounting for the number of traits tested ($P < 0.05/245$), we identified eight traits significantly correlated with FG or RG. We found a stronger genetic correlation between FG and T2D ($r_{g,FG-T2D} = 0.56$, s.e. 0.07; $P = 7.3 \times 10^{-17}$) than between RG and T2D ($r_{g,RG-T2D} = 0.32$, s.e. 0.04; $P = 2.7 \times 10^{-14}$, Fig. 2, Supplementary Data 8). This observation is consistent with previous studies showing a partial, yet significant, genetic overlap between glycemic traits and liability to T2D[9]. Moreover, waist circumference and birth weight (BW) adjusted for maternal genotype were genetically correlated with FG but only marginally with RG ($r_{g,FG-Waist} = 0.29$, s.e. 0.07; $P = 1.5 \times 10^{-5}$; $r_{g,FG-BW} = -0.31$, s.e. 0.07; $P = 3.6 \times 10^{-5}$), while the opposite pattern was observed for heart rate (HR; $r_{g,RG-HR} = 0.29$, s.e. 0.07; $P = 1.5 \times 10^{-5}$; Fig. 2, Supplementary Data 8).

### GWAS meta-analysis of RG and FG in 518,615 individuals

To further improve the statistical power for discovering blood glucose-associated loci, we meta-analysed our UKB meta-GWAS of RG with a large published GWAS of FG from the MAGIC consortium ($N = 151,188$, with summary statistics for 6,094,831 SNPs)[12], thereby reaching a total sample size of 518,615. The LDSC $r_g$ estimates between these two GWAS (FG vs. RG meta-GWAS) is $r_g = 0.82$ (s.e = 0.05). Therefore, with an increased accuracy to estimate $r_g$, this analysis reveals a small but significant differential genetic control between RG and FG. We also detected significant heterogeneity of SNP effects ($P_{HET} < 5 \times 10^{-8}$) between these two GWAS at 11 FG-associated loci, seven of which did not reach genome-wide significant association with RG (Supplementary Data 9). Among the other four loci significantly associated with both FG and RG, rs13431652 (near G6PC2) and rs1604038 (near SLC2A2) showed consistent but stronger effect on RG than on FG, while the effect size at the ABCB11 intronic variant rs853777 was larger on FG than RG (Supplementary Data 9). Interestingly, the MTNR1B intronic variant rs10830963 showed a significant oppositive effect on FG

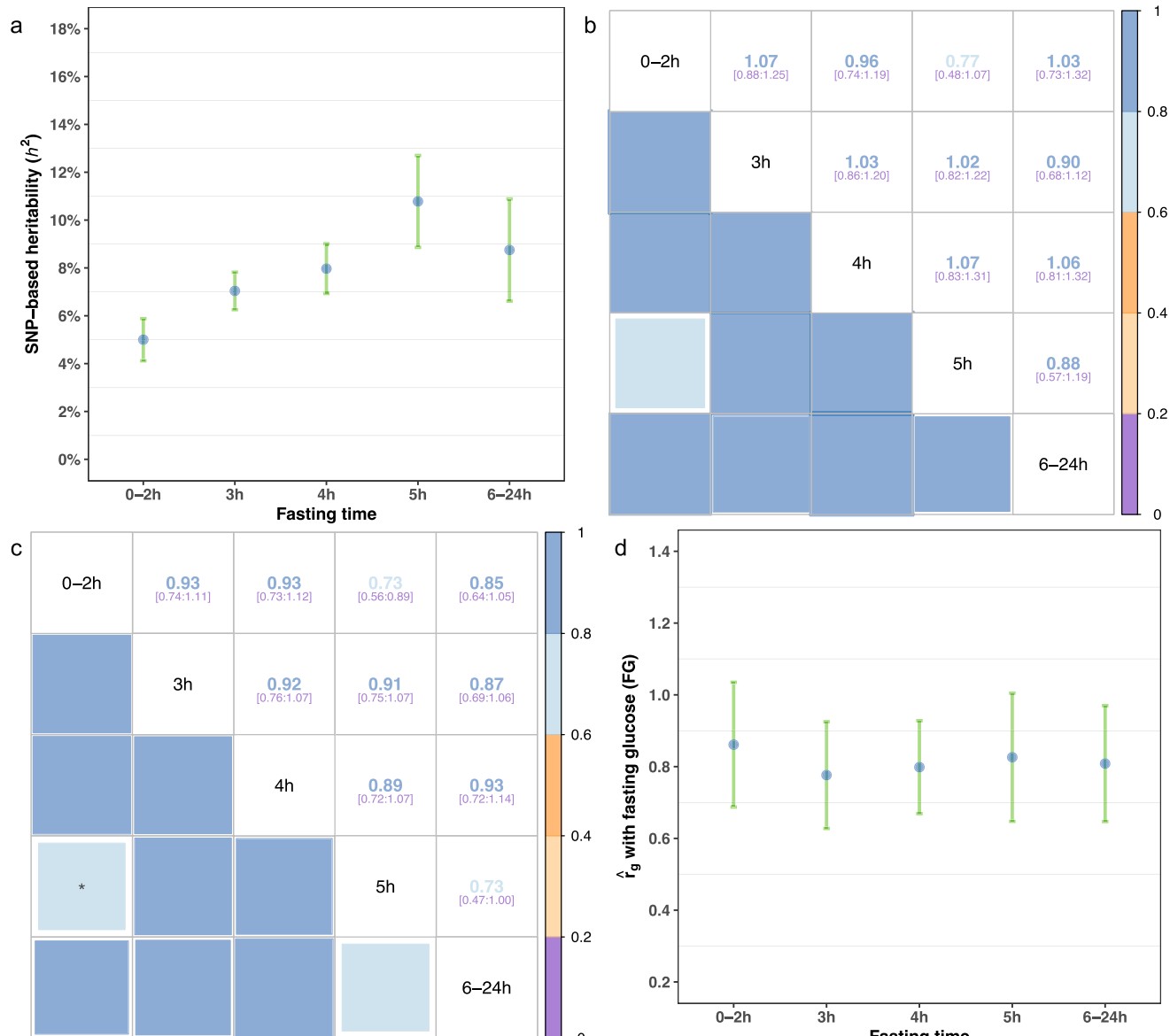

**Fig. 1 | Estimation of time-dependent genetic architecture parameters for random glucose in UKB. a** SNP-based Heritability ($h^2$) estimates for each subgroup, obtained through Haseman-Elston regression analysis based on all samples, are presented with 95% confidence intervals (95% CIs). **b** Genetic correlation estimates (and their 95% CIs) between pairwise subgroups obtained through LDSC analysis. **c** Genetic correlation estimates (and their 95% CIs) between pairwise subgroups obtained through Haseman-Elston regression analysis. The only pair of subgroups with a genetic correlation significantly lower than 1 after Bonferroni correction is marked with an asterisk. **d** Genetic correlations between fasting glucose (Lagou et al. 2021) and glucose levels in each subgroup, as estimated through LDSC analysis. Error bars in this panel represent 95% confidence intervals (95% CIs). Of note, LDSC $r_g$ estimates beyond the upper bound (> 1) are due to sampling variations. The number of samples used to compute the SNP-based heritability (**a**) and genetic correlations (**d**) for each subgroup are as follows: 0-2 h (95,199), 3 h (108,222), 4 h (80,645), 5 h (43,938) and 6-24 h (39,423). All SNP-based heritability estimates and genetic correlations in this figure can also be found in tabular form in Table 1 and Supplementary Data 3-4.

($\beta$-0.08 mmol/L per G allele; $P = 1.24 \times 10^{-211}$) versus RG ($\beta$--0.01 mmol/L per G allele; $P = 2.32 \times 10^{-9}$).

In total, we identified 156 COJO SNPs from the meta-analysis of FG and RG GWAS. This represents 90 more signals than the 66 COJO SNPs identified from re-analysing FG GWAS summary statistics of Lagou et al. (2021) alone (Table 2, Supplementary Fig. 3c). Next, we sought to evaluate if these additional SNPs improve the prediction accuracy of FG in an independent sample of 13,781 unrelated European ancestry participants from the Lifelines cohort study[25] without diabetes. We measured prediction accuracy as the squared correlation between FG and polygenic scores (PGSs) based either on COJO SNPs alone or on 1.1 M HapMap 3 SNPs (Methods). Genome-wide PGS predictors using HapMap 3 SNPs were obtained using the SBayesR method[26]. Overall,

we found that combining FG and RG GWAS data leads to identifying SNPs with an increased predictive power for FG. More precisely, the prediction accuracy for FG increased from 1.61% when using 66 COJO SNPs from Lagou et al. up to 3.08% (i.e., between 1/5th and 1/3rd of the SNP-based heritability, Supplementary Data 10) when using the 156 SNPs identified in this study. Consistently, the accuracy of the SBayesR PGS also increased from 2.28% up to 4.32% (Fig. 3a, Table 2).

We also quantified the ability of our FG and RG PGSs to predict T2D in an independent sample of 6,905 cases and 46,983 controls from the Genetic Epidemiology Research on Adult Health and Aging (GERA) study cohort[27]. For these analyses, we used the area under the receiver operator characteristic curve (AUC) as a measure of prediction accuracy. The glucose PGS based on 156 COJO SNPs identified

**Table 1 | Genetic correlation estimates ($r_g$) between pairwise (fasting) subgroups, calculated using Haseman-Elston (HE) regression analysis or LDSC analysis on unrelated samples only**

| | | HE regression | | | | LDSC | | | |
|---|---|---|---|---|---|---|---|---|---|
| | | $r_g$ | s.e. | Low 95%CI[a] | High 95% CI | $r_g$ | s.e. | Low 95%CI | High 95% CI |
| *0-2 h* | *3 h* | 0.928 | 0.095 | 0.743 | 1.114 | 1.067 | 0.094 | 0.883 | 1.251 |
| *0-2 h* | *4 h* | 0.927 | 0.099 | 0.733 | 1.122 | 0.965 | 0.115 | 0.740 | 1.190 |
| *0-2h*[b] | *5 h* | 0.728 | 0.084 | 0.564 | 0.893 | 0.774 | 0.150 | 0.480 | 1.068 |
| *0-2 h* | *6-24 h* | 0.846 | 0.107 | 0.637 | 1.055 | 1.027 | 0.151 | 0.732 | 1.322 |
| *3 h* | *4 h* | 0.916 | 0.077 | 0.765 | 1.068 | 1.031 | 0.088 | 0.858 | 1.203 |
| *3 h* | *5 h* | 0.909 | 0.080 | 0.753 | 1.066 | 1.020 | 0.104 | 0.816 | 1.223 |
| *3 h* | *6-24 h* | 0.872 | 0.095 | 0.686 | 1.057 | 0.903 | 0.112 | 0.684 | 1.123 |
| *4 h* | *5 h* | 0.894 | 0.090 | 0.717 | 1.071 | 1.073 | 0.123 | 0.833 | 1.314 |
| *4 h* | *6-24 h* | 0.927 | 0.108 | 0.716 | 1.139 | 1.064 | 0.130 | 0.809 | 1.319 |
| *5 h* | *6-24 h* | 0.734 | 0.136 | 0.467 | 1.002 | 0.878 | 0.157 | 0.571 | 1.185 |

[a] 95% CI: 95% confidence interval. s.e.: standard errors.
[b] The only pair of subgroups with a genetic correlation significantly (two-sided *P*-value <0.05) different from 1 (and by HE regression method only).

**Table 2 | Summary of analyses performed on the four sets of GWAS summary data**

| | FG | mega-RG | meta-RG | meta-analysis of glucose |
|---|---|---|---|---|
| Sample size | 151,188 | 367,427 | 367,427 | 518,615 |
| Number of SNPs | 6,094,831 | 8,546,067 | 8,546,067 | 6,094,831 |
| Mean $\chi^2$ | 1.21 | 1.46 | 1.53 | 1.64 |
| Univariate LDSC intercept (s.e.) | 1.004 (0.008) | 1.043 (0.011) | 1.043 (0.012) | 1.043 (0.012) |
| Number of COJO SNPs | 66 | 109 | 127 | 156 |
| Number of clumping SNPs | 53 | 129 | 143 | 158 |
| SNP-based Heritability ($h^2$) | 11.61% (1.74%) | 6.15% (0.99%) | 6.75% (1.09%) | 6.73% (0.97%) |
| $r_g$ with FG | - | 0.809 (0.049) | 0.820 (0.052) | 0.986 (0.024) |
| **Prediction accuracy in Lifelines (% of FG variance explained by PGS)** | | | | |
| Predictor based on COJO (s.e.) | 1.61% (0.22%) | 2.54% (0.27%) | 2.78% (0.28%) | 3.08% (0.29%) |
| Predictor based on SBayesR (s.e.) | 2.28% (0.24%) | 3.91% (0.33%) | 3.88% (0.33%) | 4.32% (0.34%) |
| **Prediction of T2D risk using glucose PGS in GERA (AUC)** | | | | |
| Predictor based on COJO (s.e.) | 0.5309 (0.0038) | 0.5547 (0.0038) | 0.5558 (0.0038) | 0.5497 (0.0038) |
| Predictor based on SBayesR (s.e.) | 0.5500 (0.0038) | 0.5684 (0.0037) | 0.5625 (0.0038) | 0.5664 (0.0037) |
| **Prediction of T2D risk using T2D PGS in GERA (AUC)** | | | | |
| Predictor based on COJO (s.e.) | 0.5876 (0.0037) | | | |
| Predictor based on SBayesR (s.e.) | 0.6269 (0.0036) | | | |
| **Prediction of T2D risk in GERA using a combined glucose and T2D PGS (AUC)** | | | | |
| Predictor based on COJO (s.e.) | 0.5893 (0.0037) | 0.5938 (0.0037) | 0.5948 (0.0037) | 0.5937 (0.0037) |
| Predictor based on SBayesR (s.e.) | 0.6295 (0.0036) | 0.6324 (0.0036) | 0.6315 (0.0036) | 0.6325 (0.0036) |

FG, summary statistics of fasting glucose from Lagou et al. (2021);
mega-RG, mega-GWAS of RG, summary statistics of random glucose modelled by mega-GWAS approach using UKB samples;
meta-RG, meta-GWAS of RG, summary statistics of random glucose modelled by meta-GWAS approach using UKB samples;
meta-analysis of glucose, summary statistics obtained through meta-analyzing FG and meta-RG.

from our meta-analysis of FG and RG showed an AUC = 0.5497 (s.e. 0.0038), that is significantly larger than AUC = 0.5309 (s.e. 0.0038) reached when only using 66 COJO SNPs from Lagou et al. ($P = 4.7 \times 10^{-4}$, Fig. 3b, Table 2, Supplementary Data 10). Similarly, the accuracy of the SBayesR PGS also increased from AUC = 0.5500 (s.e. 0.0038) to AUC = 0.5664 (s.e. 0.0037) ($P = 2.0 \times 10^{-3}$, Fig. 3b, Table 2). Note that the accuracy of all glucose PGSs remained smaller than that of T2D PGSs derived from GWAS summary statistics of Xue et al.[1] (Table 2), which can be expected because of the relatively low genetic correlation between glucose and T2D, and the fact that prediction accuracy (on the liability scale) for correlated traits scales with the square of the genetic correlation[28]. Finally, we found that PGSs combining information from glucose and T2D (the benchmark measure, Methods) improved T2D discrimination in our GERA sample (Best AUC = 0.6325; s.e. 0.0036; Table 2).

**Prioritisation of glucose-related genes and pathways**

We performed a summary data-based mendelian randomization (SMR) analysis[29] to prioritise genes for which mRNA expression could mediate associations between SNPs and glucose. For these analyses, we used multi-tissue expression quantitative trait loci (eQTLs) identified in the eQTLGen study[30] ($N = 31,684$ whole blood samples), the GTEx study[31] ($N = 838$, across 49 different tissues) and the InsPIRE study[32] ($N = 420$ pancreatic islets samples). Using GWAS data from our largest meta-analysis of glucose ($N = 519$ K), we prioritised 185 genes passing both the SMR and Heterogeneity In Dependent Instruments (HEIDI) tests ($P_{SMR} < 3.20 \times 10^{-6} = 0.05/15,645$ and $P_{HEIDI} > 0.01$; Methods), suggesting increased evidence of a pleiotropic or a causal effect of these genes on glucose levels. A complete list of these 185 genes (hereafter referred to as SMR genes) is provided in Supplementary Data 11. This list includes *GCK, NFX1, CGREF1, CCNE2, QPCTL, ABHD1, SLC39A13*,

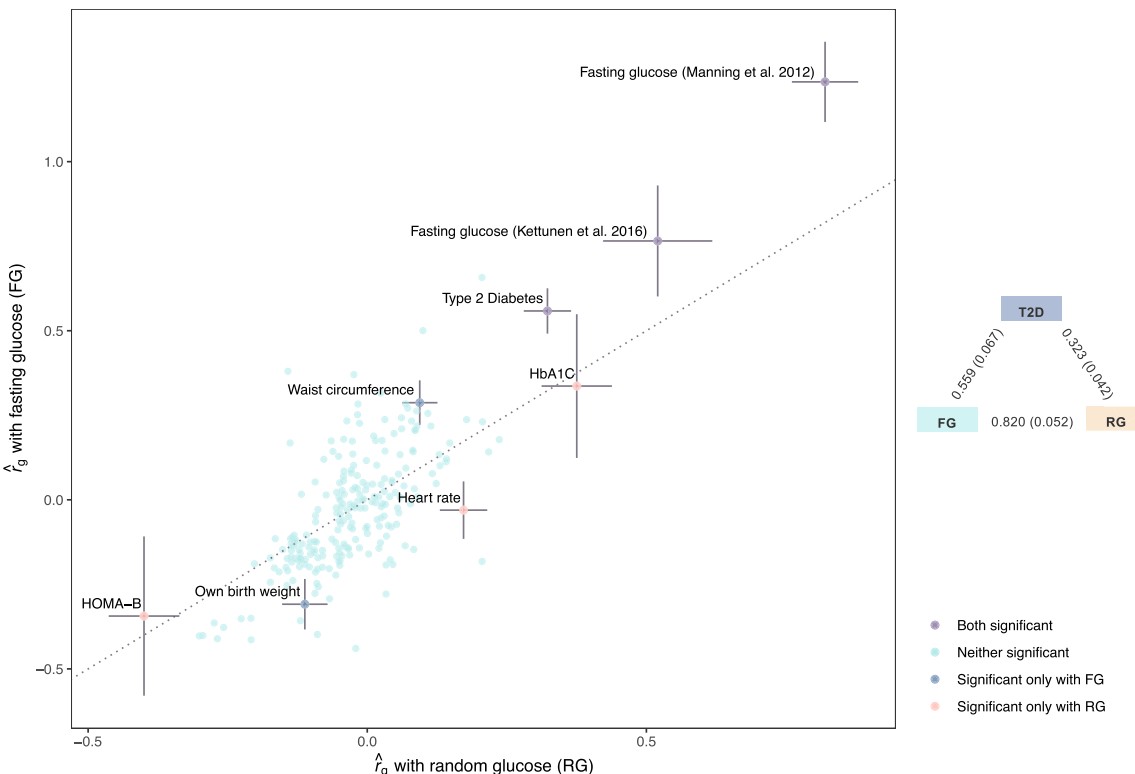

**Fig. 2 | Estimates of genetic correlation between random glucose/fasting glucose and 245 traits in LD Hub.** The genome-wide genetic correlations were estimated using LDSC regression. The x-axis represents $r_g$ estimates obtained through meta-GWAS of RG in UKB, and the y-axis represents $r_g$ estimates obtained through GWAS summary statistics of FG from Lagou et al. (2021). Of the traits analyzed, eight showed $r_g$ estimates with either RG or FG that passed the multiple testing significance threshold (two-sided $P$-value < 0.05/245, or a Bonferroni corrected $P$- value threshold of $P$ < 0.05 adjusted for 245 tests) and were annotated on the figure. The dots are colored according to the level of significance with both glucose traits. Error bars are standard errors (*s.e.*) of $r_g$ estimates. The triangle plot on the right panel shows the $r_g$ between RG, FG, and T2D. RG is from the meta-GWAS in UKB (current study), FG is Lagou et al. 2021, and T2D is from Mahajan et al. 2018. Of note, the inflation in $r_g$ estimates (> 1) is due to large sample overlaps between the studies (i.e., the same study cohort being used in multiple studies).

*SLC12A4, YWHAB, ACVR1C, TRIM59, ITFG3, SMC4, INTS8, TP53INP1, ZCWPW1, KLHL42* and *SYNM* previously associated with glucose measurements, insulin measurements and T2D. Another number of SMR genes had no prior evidence of any role in glucose metabolism, but have been implicated in glucose regulation. Those include *HBM, CREB3L4, NPEPPS, HEXIM2, LCAT, UNC13D, CHMP4B, MTMR3, RCCD1,* as well as several long non-coding RNAs and pseudogenes (Supplementary Data 11). Among those genes, *CREB3L4* was reported to negatively regulate adipogenesis when expressed in adipose tissues[33]. Importantly, adipose tissues are involved in insulin resistance and T2D through adipokines secretion affect systemic glucose homeostasis[34]. Therefore, differential expression of *CREB3L4* may change adipokine profile and, thereby, alter insulin sensitivity (Supplementary Fig. 4).

Next, we compared our SMR results across tissues and found that 108/185 (i.e., ~58%) SMR genes had a significant effect size in at least two tissues. The remaining 77 SMR genes were more often associated with expression in pancreatic islets (16 genes), blood (15 genes) and testis (10 genes), although this enrichment was not statistically significant (Fisher Exact Test $P$ > 0.7). We then focused on the 108 SMR genes detected in at least two tissues and quantified the heterogeneity of estimated effect size of gene expression on glucose levels. Overall, we found consistent effect sizes of gene expression across tissues (median Cochran's heterogeneity $I^2$ ~ 40%). However, we also observed 12/108 genes (*KLHL42, STEAP1, MBTPS1, TAPBP, TP53INP1, SMC4, TMEM45A, PLEKHM1, CCNE2, ZCWPW1, PABPC1L, YWHAB*) for which the estimated effects had inconsistent direction across tissues. For instance, *TMEM45A* expression in pancreas, pancreatic islets, pituitary, spleen and blood was positively associated with glucose, while expression in omentum, artery (aorta and tibial), spinal cord and cultured fibroblast was negatively associated with glucose. This pattern can be explained by various mechanisms causing differential regulation of gene expression across tissues including the fact that eQTLs can have opposite effects across tissues as reported previously[31]. For example, the G allele at rs4132537, an eQTL for *TMEM45A* expression, has opposite effects on gene expression in arteries and pituitary. Besides these 12 genes displaying heterogeneous effects on glucose, *GCK* showed the largest coefficient of variation of effect sizes across 7 tissues although estimates were consistently positive (Supplementary Data 11). Importantly, *GCK* also had the largest effect size on glucose levels ($b_{SMR}$ = 0.27 mmol/L per SD of *GCK* expression in blood; s.e. = 0.03; $P_{SMR}$ = 1.6 × 10$^{-19}$, $P_{HEIDI}$ = 0.012), consistent with its glucose sensing role[35].

Finally, we used the *GENE2FUNC* module of the online FUMA GWAS platform[36] to annotate SMR genes with biological and functional information. Overall, we found that SMR genes are down-regulated in liver, muscle, pancreas, heart and kidney (Supplementary Fig. 5), and significantly (False discovery rate < 5%) enriched among genes involved in peptidase activity (Supplementary Fig. 6A), vacuole and endoplasmic reticulum membrane organisation (Supplementary Fig. 6B and 6C). Moreover, SMR genes were significantly enriched among genes involved in myogenesis, MTORC1 signalling, as well as within pathways related to myometrial relaxation/contraction and G-protein-coupled receptors (in particular class B secretin-like family) activity (Supplementary Fig. 7). We also compared biological and functional enrichments of SMR genes identified through analyses of FG ($N$ = 23 genes; Supplementary Data 11) and our meta-RG ($N$ = 148 genes;

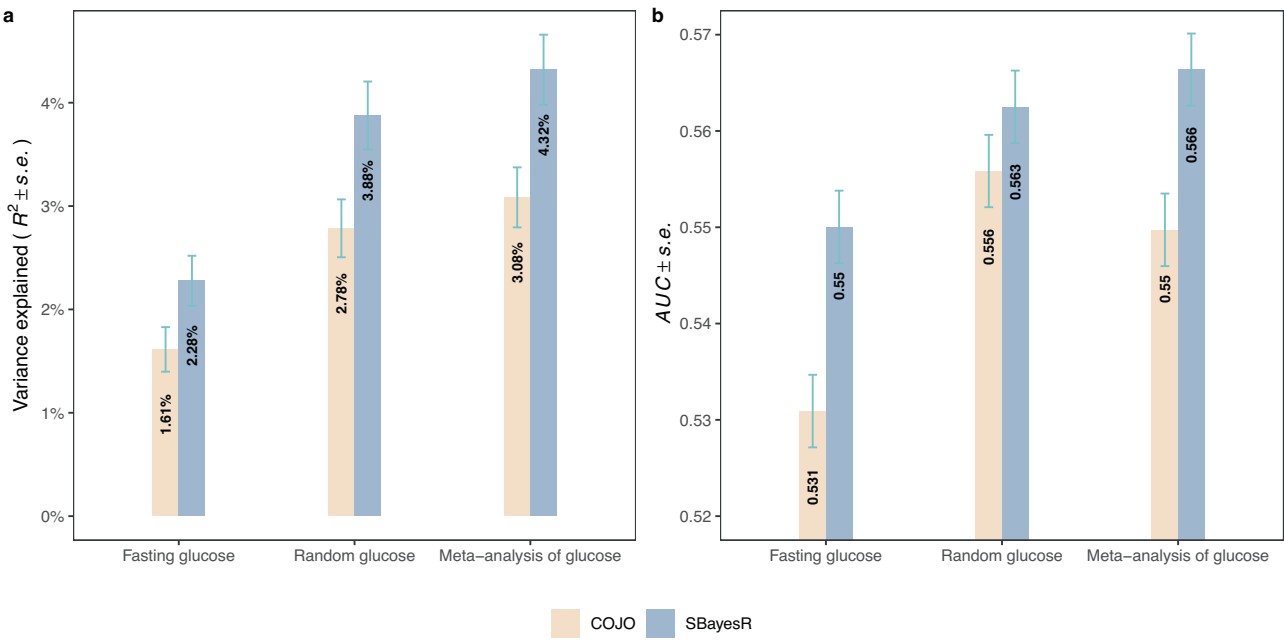

**Fig. 3 | Polygenic prediction of Fasting Glucose.** Polygenic scores were constructed using either GCTA-COJO genome-wide significant SNPs or SBayesR methods, and the standard errors (*s.e.*) were calculated using the Jackknife method. **a** Prediction $R^2$ when predicting fasting glucose in unrelated, nondiabetic Lifelines samples ($N = 13,781$). **b** Area under the receiver operator characteristic (ROC) curve (AUC) when predicting T2D risks in unrelated GERA samples (6905 T2D cases and 46,983 controls). The x-axis indicates the discovery samples used to generate the polygenic scores, while the y-axis represents the variance explained by the polygenic score (**a**) or the prediction accuracy (i.e., AUC metric) of the polygenic score (**b**). Error bars in both panels represent standard errors (*s.e.*). Fasting glucose is the GWAS summary statistics of FG from Lagou et al. (2021), random glucose is from the meta-GWAS of RG from the current study, and meta-analysis of glucose refers to the summary statistics from meta-analysis of meta-GWAS of RG and FG.

Supplementary Data 11) GWAS, but did not find a significant differential enrichment between these two sets of genes.

In summary, we prioritise here 185 genes whose expression across multiple tissues may explain how SNPs can induce physiological glucose variation. Further functional experiments are required to fully characterise a potential causal relationship between steady state gene expression of these genes and glucose levels.

### Missing heritability and mapping of future discoveries

Finally, we quantified the enrichment of SNP-based heritability of FG in genomic regions near the 156 COJO SNPs, by partitioning the genome into regions spanning the COJO SNPs vs the rest. Genome-wide significant loci were defined as genomic segments centred around each of these 156 COJO SNPs and including all SNPs within a certain window. We varied the window size from 10 kb up to 1 Mb.

We found that although these 156 SNPs only explain ~3% of FG variance in our Lifelines sample, ~50% of the FG SNP-based heritability, i.e., ~10% of FG variance, can be explained by SNPs in the close vicinity of the 156 COJO SNPs identified in our largest meta-analysis (Fig. 4). Note that the SNP-based heritability of FG in the Lifelines sample is ~20%, that is larger than previously reported[37,38] and estimated in the UK Biobank. Nevertheless, this analysis suggests that additional associations, accounting for the difference between the ~10% of FG variance expected and the ~3% already explained, remain to be discovered within 1 Mb of the 156 COJO SNPs identified in this study.

### Discussion

In this study, we demonstrate that the genetic control of blood glucose is only marginally affected by fasting status. This implies that genetic studies aiming at detecting SNPs associated with fasting glucose may conveniently utilise routinely-collected non-fasting measures in large numbers of individuals to improve statistical power. However, such a strategy is limited by the genetic correlation between FG and RG ($r_{g,FG-RG}$-0.8), which implies, in a worst-case scenario, that SNPs detected by larger GWAS of RG would only explain up to 80% of the SNP-based heritability of FG, i.e., $0.8 \times 0.11$-0.09. GWAS power to detect associations with FG can be further improved using statistical methods integrating GWAS data from genetically correlated traits like T2D or HbA1c[39]. Besides, we highlight large differences (up to ~2-fold) in SNP-based heritability estimates between fasting times. Such differences imply that glucose levels measured after short fasting durations (e.g., within 2 h post-meal) are less informative for GWAS than those collected after long fasting times. Therefore, because they do not give the same weight to each sample, meta-GWAS strategies can yield significant improvements in statistical power (here > 15%) relative to mega-GWAS strategies implemented in previous GWAS of RG (Sinnot-Armstrong et al.[33] and Lagou et al.[34]). Interestingly, glucose was the only biomarker measured in the UKB showing such a pattern (Supplementary Data 12), suggesting that meta- and mega-GWAS approaches would be equivalent for these other biomarkers.

We report a fasting-time-dependent effect of rs1881415 (*C2CD4A/C2CD4B* locus) and rs10830963 (*MTNR1B* locus) on glucose levels. The fasting-time-dependent effect of rs1881415 is consistent with previous studies showing opposite effects of the T2D risk allele at this locus on FG and 2 h glucose[22]. This observation suggests that the association of rs1881415 with glucose and T2D could be mediated by insulin secretion and not by insulin sensitivity[24,40,41]. Interestingly, the interaction between fasting time and rs10830963 was not reported before and therefore deserve further investigation and replication in an independent sample.

Our GWAS meta-analysis of FG and RG in 518,615 individuals identified 156 common SNPs (i.e., with a Minor Allele Frequency (MAF) > 1%) associated with glucose, which cumulatively explain ~3% of FG variance. Importantly, we also derived a genome-wide predictor of glucose, with an accuracy of 4.32%, which is unprecedented for a trait like FG. Finally, we showed that ~50% of FG SNP-based heritability is

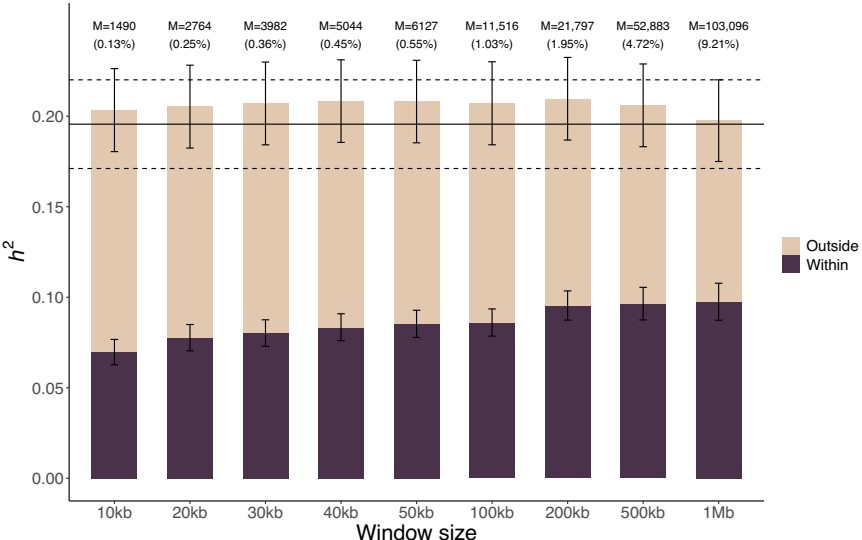

**Fig. 4 | Partitioned SNP-based heritability of FG in the vicinity of 156 glucose-associated SNPs.** Estimates of SNP-based heritability ($h^2$) were obtained in the Lifelines sample ($N = 13,781$) by partitioning the genome into SNPs within genome-wide significant loci vs. SNPs outside of genome-wide significant loci. Error bars are standard errors (s.e.). Horizontal lines (dotted = estimate ± standard error) represent the non-stratified genome-wide estimate of FG SNP-based heritability. Genome-wide significant loci were defined as genomic segments centred around each of these 156 COJO SNPs and including all SNPs within a window size varied from 10 kb up to 1 Mb. M (%) denotes the number (percentage) of SNPs within genome-wide significant loci.

concentrated within < 1 Mb of these 156 glucose-associated SNPs, suggesting a significant allelic heterogeneity at these 156 loci and that future GWAS of glucose are likely to detect new associations nearby those identified here.

Our study has a few limitations. First, we used self-reported fasting times, which may not provide a reliable assessment of actual fasting duration. Secondly, by choosing to analyse groups with at least 30,000 individuals, participants within 2h-post meal were aggregated in a single group despite the substantial changes in glucose levels occurring in that critical window (Supplementary Fig. 8). Although the vast majority (~82%) of participants in that group reported their last meal exactly 2 h prior to the assessment, we sought to quantify the genetic correlation of glucose levels between the 0–1 h ($N = 13,325$) and the 2 h group ($N = 59,494$). We used GCTA instead of LDSC to estimate this genetic correlation because the former yields more precise estimates with smaller sample sizes[42]. While standard errors remain large, we found a genetic correlation $r_g = 0.60$ (s.e. 0.17; 95%CI: 0.26–0.93) significantly lower than 1 ($P = 0.018$), suggesting some heterogeneity of SNP effects between these two sub-groups. However, there was no significant heritability difference between these groups (0–1 h: $h^2 = 0.072$, s.e. 0.026; 2 h: $h^2 = 0.066$, s.e. 0.007) implying that splitting the 0–2 h group in two sub-groups may not further improve statistical power of our meta-GWAS. Nevertheless, if sample size is large enough, more loci showing heterogeneity of SNP effects on glucose levels can be detected within that time interval. Thirdly, our GWAS of glucose were restricted to individuals with European ancestries, and therefore the transferability of our findings across ancestries is not warranted. Lastly, we did not perform any post-GWAS in vitro or in vivo studies, which can provide valuable evidence to support a role of newly discovered GWAS variants and genes in glucose homeostasis.

Altogether and despite these limitations, our study provides a strong proof-of-concept of the utility of non-fasting endpoints for genetic studies of glucose, thereby offering new opportunities for future discoveries across biobanks.

## Methods
### The UK Biobank cohort
The UK Biobank (UKB) is a large population-based prospective study with deep genetic and phenotypic data collected from over 500,000 participants recruited when aged between 40 and 69 years old. After the baseline assessment, approximately 20,000 participants attended a repeat assessment visit in 2012–2013. The North West Multi-Centre Research Ethics Committee granted ethics approval, and written informed consent was provided by all UKB participants[18].

**Genotype and quality control.** Genome-wide genotyping was performed on all UKB participants using two arrays: the UK BiLEVE Axiom Array by Affymetrix and the UK Biobank Axiom Array. Over 800,000 variants were directly genotyped, quality checked, and then imputed to the Haplotype Reference Consortium (HRC)[43] and the UK10K[44] reference panel by the UKB team[18]. We converted imputed dosage data to hard-call genotypes using PLINK[45] (v2.00aLM,--hard-call 0.1). We excluded variants with imputation score < 0.3, minor allele count (MAC) < 5, genotype missingness > 0.05 or Hardy-Weinberg equilibrium test $P$-value < $1 \times 10^{-5}$. We further restricted our analyses to 8,546,067 SNPs autosomal imputed variants with MAF ≥ 0.01.

The approach to determine ancestry of UKB samples was extensively described in a previous study[46]. Briefly, the UKB samples were projected onto the first two principal components (PCs) from the 1KG reference panel using common SNPs with MAF > 0.01 in both datasets. Individuals with posterior probability > 0.9 of belonging to the European cluster were assigned to European ancestry[47]. We further identified a subset of unrelated European individuals by constructing genomic relationship matrix (GRM) using ~1.1 M HapMap3 SNPs with a cut-off genomic relationship value of 0.05 (gcta--grm-single 0.05, gcta v1.93.1.beta)[48].

**Random glucose and other biochemical markers.** UKB performed laboratory testing on a wide range of biomarkers in serum and urine frequently measured in clinical settings to diagnose and monitor chronic disease conditions. These biomarkers were collected from all participants at the baseline assessment and those who attended the repeat assessment. Glucose was measured at serum level in mmol/L (by hexokinase analysis on a Beckman Coulter AU5800) and HbA1c was measured using (packed) red blood cell samples in mmol/mol. Following previous MAGIC's efforts, we used untransformed glucose levels in our analyses.

Similar to previous blood glucose studies[6,9,10,12,13], individuals were excluded if they were diagnosed within any subtypes of diabetes, on diabetes medications, had abnormal glucose (≤3 mmol/L or ≥11.1 mmol/L) or glycated haemoglobin (HbA1c ≥ 48 mmol/mol) levels in any visits to the assessment centre. For participants who had more than one glucose measurement, we generated the phenotype and covariates based on the records from the baseline assessment. Fasting time (i.e., how many hours since last consumption of meal or drink except for plain water; UKB Data-Field 74) ranging between 0 and 72 h across UKB samples was recorded when blood samples were taken, and we excluded the samples who reported a fasting time greater than 24 h to remove outliers and avoid recall bias ($N = 25$). After quality control, we retained 367,427 individuals of European ancestry with non-missing genotype, phenotype and covariates data available for our main analyses, of whom 280,962 were unrelated (Supplementary Methods).

### Genome-wide association studies (GWAS)

The genome-wide association analyses reported as our main GWAS result were performed using the BOLT-LMM software (v2.3.4)[49] with age (during assessment), sex, genotyping batch, assessment centre and the first 40 genetic principal components (PCs) fitted as default covariates. We used a set of 711,933 LD pruned autosomal HapMap3 SNPs (LD $r^2 > 0.9$, MAF > 0.01) as model SNPs in the analysis to correct for confounding effects (e.g., population stratification), as required by the BOLT-LMM software (v2.3.4).

We performed association analyses on glucose measurements in each of the five subgroups with the default covariates, as well as in all individuals with further adjustment of fasting time as a categorical covariate (Supplementary Data 2). We refer to the latter analysis as the mega-GWAS approach. We performed an inverse-variance weighted (IVW) meta-analysis of the GWAS results from the five subgroups and referred to it as meta-GWAS approach. We defined genome-wide significant (GWS) SNPs with a significance threshold of $5 \times 10^{-8}$ for each GWAS.

### GCTA-COJO analysis

To identify independent GWS signals, we conducted a conditional and joint (COJO) analysis of GWAS summary statistics which utilises LD information from an external reference panel to identify jointly associated signals by fitting multiple variants simultaneously in the model[21]. We used a random subset of 20,000 unrelated European ancestry samples from the UKB as LD reference. Analyses were performed assuming that SNPs 10 Mb apart or on different chromosomes were not in LD (default settings in GCTA-COJO, v1.93.1.beta).

### Estimation of SNP-based heritability and genetic correlation

We used LD score regression (LDSC)[20] and Haseman-Elston regression (HE) implemented in GCTA to estimate the SNP-based heritability of RG (e.g., in each subgroup) and genetic correlations between glucose and other traits. The LDSC intercept approximates the mean χ2 association statistic at SNPs not associated with the trait, and therefore provides a quantification of confounding due to population stratification[20]. HE regression was used whenever individual-level data were available.

We performed a range of genetic correlation analyses in this study: (1) estimated the $r_g$ of glucose levels between pairs of subgroups; (2) estimated the $r_g$ between FG and RG, using summary statistics obtained from both mega-GWAS and meta-GWAS approaches. The first set of analyses aimed to detect heterogeneity in genetic effects between different subgroups, for example the glucose levels measured at different fasting times might be under different genetic control. The second set was applied to uncover the genetic relationship between the two glucose phenotypes.

In addition, to compare the genetic correlation between FG/RG and other complex traits and diseases, we applied bivariate LDSC

implemented in the LD Hub online tool[24], which is a centralized database of summary-level GWAS results developed for screening hundreds of different complex traits and diseases for genetic correlations with a trait of interest. We included 245 GWAS summary statistics of different phenotypes available in LD Hub and compared their $r_g$ estimates with FG against those with RG. Because the T2D summary statistics recorded in LD Hub is out of date and the sample size is too small (published in 2012, $N = 69,033$)[6], we used instead the more recent and larger GWAS of T2D (published in 2018, $N = 898,130$)[5]. Significant genetic correlations were defined at the Bonferroni-corrected $P$-value $< 2.04 \times 10^{-4}$ (0.05/245).

### Meta-analysis

Fixed-effect inverse variance weighted meta-analyses were conducted using METAL (version 2011-03-25) (https://genome.sph.umich.edu/wiki/METAL)[50] and reported as the main results in this study. Before meta-analysis, we checked the genetic heterogeneity and sample overlap between or among GWAS summary results and only retained SNPs that were common to all data sets. We performed meta-analysis of the GWAS results across five subgroups in UKB for 8,546,067 common SNPs. Heterogeneity of allelic effects among subgroups were evaluated by the Cochran's Q test[51] implemented in the METAL package.

We also conducted meta-analysis of GWAS results from our meta-GWAS of RG (within the UKB) and a GWAS of FG from the MAGIC consortium[12] (i.e., metaGlu). Summary statistics of the FG GWAS from the MAGIC consortium[12] were imputed to all-ancestries 1000 Genome reference panel and available for 8,658,737 SNPs. We then filtered out SNPs which reported pairs of alleles did not match the pairs of alleles in the UKB. Since it was difficult to determine the minor allele for which the minor allele frequency (MAF) reported, we imputed the allele frequencies (AFs) for SNPs common to the two data sets (i.e., a total of 6,094,831 SNPs) based on the AFs of UKB SNPs to avoid ambiguous determination of the minor allele.

### Summary-data-based Mendelian Randomization (SMR) analysis

We performed an SMR analysis[29] to prioritise putative causal genes underlying glucose phenotypes. SMR integrates summary statistics from GWAS and expression quantitative trait loci (eQTL) studies and performs a Mendelian Randomization using the top associated cis-eQTL of a gene (the most significant SNP associated with the expression of this gene within ±1 Mb) as an instrumental variable for the gene-trait association. Genes that are SMR significant show evidence for a causal relationship with the trait mediated through gene expression. A subsequent test, the HEterogeneity In Dependent Instruments (HEIDI) test is used to increase the likelihood that our findings reflect causality or pleiotropy over the possibility that causal variants for glucose and that for gene expression are in LD. We used eQTL summary data from the eQTLGen study[30] ($N = 31,684$ whole blood samples), the GTEx study[31] ($N = 838$, across 49 different tissues) and the InsPIRE study[32] ($N = 420$ pancreatic islets samples), most of which were derived from glucose relevant tissues. We focused our analyses on gene probes with at least one genome-wide significant eQTL ($P_{eQTL} < 5 \times 10^{-8}$). SMR significant results were declared at $P_{SMR} < 0.05/m$, where $m$-15,645 corresponds to $3.2 \times 10^{-6}$. The significance level of HEIDI test was set at $P_{HEIDI} > 0.01$ as recommended by Wu et al. (2018)[52].

The SMR prioritized genes were then taken forward to the GENE2FUNC platform of FUMA GWAS ("Functional Mapping and Annotation of Genome-Wide Association Studies")[36] to gain insights into putative biological mechanisms. To be specific, tissue specificity of these prioritized genes was provided by evaluating their over-representation in sets of differentially expressed genes (DEGs) for each of the 30 general tissue types based on GTEx v8 RNA-seq data[31]. Besides, enrichment of these prioritized genes in biological pathways and functional categories was accessed using the hypergeometric tests

by testing them against gene sets obtained from MsigDB[53] and WikiPathways[54]. Multiple testing correction (i.e., Benjamini-Hochberg by default) was performed per data source of tested gene sets (e.g., GO biological processes, hallmark genes). FUMA reported all gene sets with adjusted $P \leq 0.05$.

## Out-of-sample polygenic prediction

We assessed the predictive ability of four polygenic scores (PGS) of glucose in two datasets independent from our discovery GWAS. PGS were derived from summary statistics of GWAS RG (mega-GWAS or meta-GWAS), FG (data from Lagou et al. (2021)[12]) as well as the meta-analysis of FG with our mega-GWAS (metaGlu).

**Prediction of fasting glucose.** We predicted fasting glucose in 13,781 unrelated participants of the Lifelines study[25,55], a multi-generational population-based cohort study initiated with a research focus of the onset and development of chronic diseases and healthy ageing. We focused our analyses on measurements recorded during the baseline assessment, which include a quantification of fasting plasma glucose using the hexokinase method. These 13,781 Lifelines participants were genotyped using the Illumina global screening array (GSA) Beadchip-24 v1.0, as part of the UMCG Genetics Lifelines Initiative (UGLI). After initial quality control, approximately 570,000 SNPs that passed QC filters were subsequently imputed using the HRC panel by the central UGLI team. We restricted our analysis to adult individuals (age >18 years) whose FG levels were lower than 7.0 mmol/L and without self-reported diabetes.

Using this set of unrelated samples ($N = 13,781$) as the target dataset, we conducted polygenic score analyses. PGS were constructed for the target samples using the SNP effects re-estimated by GCTA-COJO and SBayesR[26] methods (PLINK v1.90b6.11 --score function), and the prediction accuracies were measured by the proportion of phenotypic variance explained by the polygenic profiles in the linear regression ($R^2$). We ran SBayesR using a banded LD matrix with a window size of 3 cM per SNP computed based on 1.1 million common HapMap 3 SNPs in 10,000 randomly selected and unrelated UKB samples.

**Genetic risk prediction of T2D.** We predicted T2D in 6,905 T2D cases and 46,983 controls from the Genetic Epidemiology Research on Adult Health and Aging (GERA) cohort[27], a large, multiethnic, and comprehensive population-based cohort with >100,000 subjects genotyped on the Affymetrix Axiom Genotyping System[56]. Detailed QC and imputation procedures on GERA have been described in our previous studies[57]. Our analysis focused on individuals of European ancestry and excluded related individuals at a genetic relatedness threshold of 0.05. After QC, 6,905 T2D cases and 46,983 controls were retained. As described above, we constructed PGS using SNP effects re-estimated by GCTA-COJO and SBayesR[26] methods, and quantified the prediction accuracy by measuring the Area Under the receiver operator characteristic (ROC) Curve (AUC) [R library pROC[58]]. As a benchmark, we also meta-analysed T2D summary statistics from DIAbetes Genetics Replication and Meta-analysis consortium (DIAGRAM, 34,840 cases and 114,981 controls)[6] and UKB (21,147 cases and 434,460 controls)[1] and used it as the discovery set (55,987 cases and 549,441 controls) to predict T2D disease risks in GERA. Description of the latter meta-analysis has been described previously[1].

## Partitioned SNP-based heritability of FG

We sought to quantify the contribution to FG glucose variance of sub-significant SNPs located within glucose-associated loci. For that, we partitioned the SNP-based heritability ($h^2$) of FG into two genomic regions (i.e., within- and outside-GWS loci region), where the GWS loci of glucose were defined as genomic segments centred around each of the 156 COJO SNPs (identified using meta-analysis of glucose) and including all SNPs within a specific window size (10 kb, 20 kb, 30 kb, 40 kb, 50 kb, 100 kb, 200 kb, 500 kb, and 1 Mb).

To do this, we first estimated the $h^2$ of FG using the same set of unrelated individuals from the Lifelines cohort ($N = 13,781$) by the genomic restricted maximum likelihood (GREML) method implemented in the GCTA software package[48]. The FG phenotype was corrected for age and age$^2$ and the first 20 PCs (calculated from HapMap3 SNPs) within each sex. Then, we stratified the variants into two bins and computed the GRMs from the variants in each of these bins, and fitted jointly in a multicomponent GREML analysis using GCTA. The proportion of FG variance explained by within- and outside- GWS loci were used to compare their relative contributions to the total FG variance explained. We repeated this procedure for all the 9 window sizes defined previously.

## Reporting summary

Further information on research design is available in the Nature Portfolio Reporting Summary linked to this article.

## Data availability

All datasets used in this study are available in the public domain. This study uses genotype and phenotype data from UK Biobank Resource under project 12505. UKB data can be accessed upon request once a research project has been submitted and approved by the UKB committee. Data on glycaemic traits were downloaded from www.magicinvestigators.org. Other datasets used in these analyses can be sourced from: eQTLGen Consortium Data, http://www.eqtlgen.org/cis-eqtls.html. GERA, https://www.ncbi.nlm.nih.gov/projects/gap/cgi-bin/study.cgi?study_id=phs000674.v2.p2. GTEx, https://gtexportal.org/home/datasets. HapMap3, https://www.sanger.ac.uk/resources/downloads/human/hapmap3.html. Lifelines cohort study, https://www.lifelines.nl/researcher. Banded LD matrix of -1.1 million HapMap3 SNPs computed from 10,000 unrelated UKB individuals of European ancestry: https://cnsgenomics.com/software/gctb/#Download.

Genome-wide association summary statistics generated from this study (i.e., mega-GWAS of random glucose, meta-GWAS of random glucose, meta-analysis of glucose) are available for download from https://cnsgenomics.com/data/qiao_et_al_2023_nc/.

## Code availability

Scripts used to perform various analyses reported in this study are publicly available an Github at https://github.com/uqzqiao/random-glucose and have been deposited at https://doi.org/10.5281/zenodo.7456276

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

## Acknowledgements

This research was supported by the Australian Research Council (DE200100425, FL180100072) and the Australian National Health and Medical Research Council (1173790 and 1113400). Any opinions, findings, and conclusions or recommendations expressed in this material are those of the authors and do not necessarily reflect the views of the funding bodies. The funders had no role in study design, data collection and analysis, decision to publish, or preparation of the manuscript. This research has been conducted using the UK Biobank Resource under project 12505. The LifeLines Cohort Study, and generation and management of GWAS genotype data for the LifeLines Cohort Study is supported by the UMCG Genetics Lifelines Initiative (UGLI), the Netherlands Organization of Scientific Research NWO (grant 175.010.2007.006), the Economic Structure Enhancing Fund (FES) of the Dutch government, the Ministry of Economic Affairs, the Ministry of Education, Culture and Science, the Ministry for Health, Welfare and Sports, the Northern Netherlands Collaboration of Provinces (SNN), the Province of Groningen, University Medical Center Groningen, the University of Groningen, Dutch Kidney Foundation and Dutch Diabetes Research Foundation. The authors wish to acknowledge the services of the Lifelines Cohort Study, the contributing research centers delivering data to Lifelines, and all the study participants. The authors are grateful to Yuanhao Yang, Yang Wu and Guiyan Ni for helpful discussions and comments on the manuscript.

## Author contributions

L.Y. and Z.Q. designed the study. Z.Q. and J.S. performed data analyses. L.Y. supervised the project. Z.Q. and L.Y. interpreted the results and wrote the manuscript. J.A.R., A.X., X.L., K.P., H.S., P.M.V., and N.R.W. contributed to reviewing and editing the manuscript. All authors read, approved, and provided feedback on the final manuscript.

## Competing interests

The authors declare no competing interests.

## Additional information

## Lifelines Cohort Study

Peter M. Visscher ⓘ [2], Xueling Lu[3,4], Katri Pärna[3,5] & Harold Snieder ⓘ [3]

A full list of members and their affiliations appear in the Supplementary Information.

