## [Peer Review File · Nature Communications]

Estimation and implications of the genetic architecture of fasting and non-fasting blood glucoseREVIEWER COMMENTS

Reviewer #1 (Remarks to the Author):

Qiao et al., report a very interesting study leveraging UKB data where mostly random glucose levels were collected to improve the current knowledge about the genetic basis of glucose levels, an important trait for glucose homeostasis. The authors provided an interesting analytical strategy making use of 5 fasting time ranges to astutely estimate genetic correlations between FG and RG from these ranges. The high genetic correlations observed, except for the first post-prandial period, encouraged performing a meta-analysis with the most recent meta-analysis dataset of FG. This analysis provided a significant increase estimated of 16% in the number of association signals detected. The global analytical work is very solid and used sound methods, although the annotation part is very superficial. The prediction of T2D is limited as well and I have few comments about these aspects in particular. Overall, I find the study very well written and the authors provide an interesting and substantial addition to the field of the genetics of fasting glucose.

1. Please comment on the rs1881415 heterogeneity association. What is the association level in the Saxena et al meta-analysis for 2h glucose post glucose challenge or any work related to post-prandial glucose levels, if any? Confirm if the direction of effect is consistent with higher risk of T2D of the allele associated with higher 3h post meal?
2. Did the authors performed prediction of T2D using the 156 COJO SNPs by considering the association with T2D? Given that many FG and RG determinants are demonstrated not to be T2D determinants, how informative one would expect this predictive value to be? Please comment on the extremely low predictive power of glucose SNPs for T2D in general.
3. Only if the data is reasonably accessible in the cohorts analyzed here, could the authors test the predictive power of glucose determinants on cardiac related events or diseases?
4. I would be curious to see if using a PGS with only those SNPs that provide consistent direction of effects on T2D risk (increasing glucose levels allele increase risk for T2D, without considering significance of course), would have an improved predictive power.
5. The authors are invited to explicit how they came with the estimation about the likelihood that a third of the future FG-associated SNPs to be discovered within 1Mb of the currently described loci obtained from this meta-analysis.
6. Could the authors confirm there are no existing eQTL datasets for a more relevant tissue to glucose homeostasis other than blood? (e.g pancreatic islets, muscle, adipose tissue, where one putative gene (CREB3L4) seem to be mentioned as likely to regulate adipogenesis). The functional annotation part could gain in interest by integrating more data.
7. Please comment on HbA1c levels availability in UKBB and if yes, consider and discuss on the interest in performing a genetic comparison with RG in the same way they compared to FG.

Reviewer #2 (Remarks to the Author):

J. Qiao and colleagues performed an exciting study on the genetic architecture of fasting and random glucose.

1-The authors may precise in the introduction that the genetic architecture of fasting glucose level and type 2 diabetes is only partially overlapping (Bouatia-Naji et al., Science 2008, Bouatia-Naji et al., Nat Genet 2009).

2-I am surprised by the huge number of related individuals excluded from the analysis (N=86465) in the UK Biobank. Many GWAS have been previously performed in the UK Biobank, and I never saw that many individuals excluded from the analysis because of relatedness issues. The authors may consider using a more relaxed threshold of relatedness (e.g. exclusion of 1st and 2nd degree relatives) to

improve statistical power. Another option can be to adjust the GWAS analysis for relatedness, as previously described in literature.

3-Related to my previous comment, did the authors apply similar exclusion procedure for related individuals in the MAGIC sample.

4-The authors defined five groups based-on their self-reported fasting time (0-2h, 3h, 4h, 5h and 6-24h) to ensure that at least 30K participants were included in each group. They may also consider physiological considerations (e.g. shape of post-prandial glucose curves) to reach an optimal definition of groups. Glucose values change a lot during the 2h postprandial period (in relation to the 1st and 2nd phase of insulin secretion), and it seems a little bit counterintuitive to have a 0-2h group.

5-The authors may precise the ancestry of the Lagou et al. sample (UK Biobank is predominantly EU).

6-For clarity, please define in the text what means LDSC intercept.

7-'Overall, we found no significant evidence of heterogeneous SNP effects across fasting time groups except for rs1881415 ($P_{HET} < 5 \times 10^{-8}$), which only showed a significant effect on glucose levels within 3h post-meal (Supplementary Table 7). Importantly, this intergenic variant is in highly LD ($r^2 > 0.8$) with SNPs previously associated with fasting proinsulin and T2D risk'. The SNP rs1881415 is also in strong LD with another SNP strongly associated with fasting glucose in the European population (Chen et al., Nat Genet 2021). This is somewhat discordant with the author's findings and should be mentioned and discussed.

8-'Note, as a benchmark, that this estimate is similar to the 170 LDSC genetic correlation between males and females for traits like body mass index (BMI) and waist circumference'. I think this sentence may be out of context. Please consider removing.

9-'We also quantified the ability of our FG and RG PGSs to predict T2D in an independent sample of 6,905 cases and 46,983 controls from the Genetic Epidemiology Research on Adult Health and Aging (GERA) study cohort'. As GERA is multiethnic, the authors may consider performing the T2D genetic prediction study on the EU subset of GERA participants for consistency, as they identified FG and RG-associated SNPs in populations of EU ancestry.

10-The authors may consider performing molecular pathway / gene function / tissue expression GWAS gene-enrichment analyses to investigate the biological mechanisms underlying the genetic architecture of FG and RG. As the genetic overlap between FG and RG is partial, they may identify different mechanisms involved in FG or RG genetic variation.

11-The authors did not perform post-GWAS in vitro / in vivo studies (e.g. cellular / animal studies) to investigate the role of newly discovered GWAS variants and genes in glucose homeostasis. I understand that this may be considered as beyond the scope of the current study, but they may at least mention this limitation in the discussion.

REVIEWER COMMENTS

Reviewer #1 (Remarks to the Author):

Qiao et al., report a very interesting study leveraging UKB data where mostly random glucose levels were collected to improve the current knowledge about the genetic basis of glucose levels, an important trait for glucose homeostasis. The authors provided an interesting analytical strategy making use of 5 fasting time ranges to astutely estimate genetic correlations between FG and RG from these ranges. The high genetic correlations observed, except for the first post-prandial period, encouraged performing a meta-analysis with the most recent meta-analysis dataset of FG. This analysis provided a significant increase estimated of 16% in the number of association signals detected. The global analytical work is very solid and used sound methods, although the annotation part is very superficial. The prediction of T2D is limited as well and I have few comments about these aspects in particular. Overall, I find the study very well written and the authors provide an interesting and substantial addition to the field of the genetics of fasting glucose.

We thank the referee for their accurate summary of our study, for highlighting many of its strengths and for making suggestions to improve it. We address below each of the referee's comments.

1. Please comment on the rs1881415 heterogeneity association. What is the association level in the Saxena et al meta-analysis for 2h glucose post glucose challenge or any work related to post-prandial glucose levels, if any? Confirm if the direction of effect is consistent with higher risk of T2D of the allele associated with higher 3h post meal?

We thank the referee for this suggestion. We looked up summary statistics for rs1881415 in the currently largest GWAS of 2h-glucose from Chen et al. (2021)¹ [max $N \sim 86K$, i.e., larger than Saxena et al. (2010)², $N < 46K$] and compared effect sizes with that on FG, HbA1c, fasting insulin (FI) and T2D. The results are summarised in the table below and in **Supplementary Table 7**.

These results show that the T allele at rs1881415 is positively associated with T2D risk^{3,4}, while showing opposite effects on 2h glucose levels (negative effect) vs FG (positive effect). Our data recapitulate this heterogeneous pattern (effect sizes are sign-consistent) although the relatively small number of UK Biobank participants fasting for > 5 hours reduces statistical power to detect the negative effect on FG at genome-wide significance. As discussed elsewhere⁵, these heterogeneous patterns of associations with various glycemic traits suggest that this locus contributes to T2D risk through beta-cell dysfunction and not through reduced insulin sensitivity.

Following the referee's comment, we have now rephrased the paragraph (in the result section) reporting a significant heterogeneous effect at rs1881415 as (Lines 140–147): "We found genome-wide significant evidence of heterogeneous SNP effects across fasting time groups for rs1881415 ($P_{HET} < 5 \times 10^{-8}$; **Supplementary Table 7**). This variant is in high LD ($r^2 > 0.8$) with SNPs previously associated with fasting glucose, fasting proinsulin, 2h-glucose¹ and T2D risk³. Importantly, the T2D risk allele at this locus shows opposite effects on FG (positive effect) vs 2h glucose levels (negative effect). Our data recapitulates this heterogeneous pattern (i.e., effect sizes are not sign-consistent across fasting time) although the relatively small number of UKB participants fasting for > 5 hours reduces the statistical power to detect a genome-wide significant effect."

Trait	Effect Allele	Other Allele	Frequency	Effect size	Standard Error	P-value	Sample Size (N)	Ref
2h Glucose	T	C	0.582	-0.0445	0.0078	1.01E-08	63,396	(Chen et al. 2021) ¹

Fasting glucose			0.0209	0.0017	1.02E-31	200,622	
Fasting insulin			-3.00E-04	0.0019	6.68E-01	151,013	
HbA1c			-1.00E-04	0.0013	7.42E-01	146,806	
T2D	0.561	0.0462	0.0065	1.10E-12	684,683	(Mahajan et al. 2022) ³	

2. Did the authors performed prediction of T2D using the 156 COJO SNPs by considering the association with T2D? Given that many FG and RG determinants are demonstrated not to be T2D determinants, how informative one would expect this predictive value to be?

Please comment on the extremely low predictive power of glucose SNPs for T2D in general.

We thank the referee for these questions. We re-assessed the accuracy to predict T2D from various glucose PGS constructed only using SNPs with consistent direction of effects on T2D risk⁶. The Table below (now included as **Supplementary Table 10**) summarises our results and shows a slight improvement of accuracy.

Glucose PGS have a low accuracy to predict T2D largely because of the relatively low genetic correlation between them ($r_g \sim 0.3$). In fact, the transferability of prediction accuracy across traits scales with the square of the genetic correlation. We have now highlighted this in the main text (Lines 210–214): **“Note that the accuracy of all glucose PGSs remained smaller than that of T2D PGSs derived from GWAS summary statistics of Xue et al.⁶ (Table 2), which can be expected because of the relatively low genetic correlation between glucose and T2D, and the fact that prediction accuracy (on the liability scale) for a correlated trait scales with the square of the genetic correlation⁷.”**

	FG	mega-RG	meta-RG	meta-analysis of glucose
Sample size	151,188	367,427	367,427	518,615
Count of SNPs				
COJO SNPs	66	109	127	156
COJO SNPs presented in T2D sumstats	63	83	102	147
COJO SNPs with same direction of effects on T2D risk	51	50	78	108
Number of SNPs re-estimated by SBayesR (SBayesR SNPs)	1,097,166	1,108,030	1,108,030	1,097,166

SBayesR SNPs presented in the sumstats	1,078,874	1,088,847	1,088,847	1,078,874
SBayesR SNPs with same direction of effects on T2D risk	540,494	532,812	529,704	548,951
Prediction of T2D risk using glucose PGS in GERA (AUC) PGS constructed using SNPs with consistent direction of effects on T2D risks				
Predictor based on COJO (s.e.)	0.5346 (0.0038)	0.5561 (0.0038)	0.5642 (0.0038)	0.5557 (0.0037)
Predictor based on SBayesR (s.e.)	0.5498 (0.0038)	0.5864 (0.0038)	0.5818 (0.0037)	0.5565 (0.0038)
Prediction of T2D risk using glucose PGS in GERA (AUC) Reported in the main text				
Predictor based on COJO (s.e.)	0.5309 (0.0038)	0.5547 (0.0038)	0.5558 (0.0038)	0.5497 (0.0038)
Predictor based on SBayesR (s.e.)	0.5500 (0.0038)	0.5684 (0.0037)	0.5625 (0.0038)	0.5664 (0.0037)

3. Only if the data is reasonably accessible in the cohorts analyzed here, could the authors test the predictive power of glucose determinants on cardiac related events or diseases?

Taking coronary artery disease (CAD, i.e., the most common type of cardiovascular disease) as an example, we estimated the genetic correlation (r_g) between CAD and the four glucose phenotypes used in our study, and we observed that the r_g between CAD and FG was higher than that between CAD and RG. Interestingly, genetic overlap between CAD and glucose phenotypes were much lower compared to that between CAD and T2D⁴ ($r_g=0.4044$, s.e. of $r_g=0.0224$). We also used glucose PGSs to predict cardiovascular disease risks among unrelated GERA samples (13,664 cases and 37,594 controls) similarly as what had been done for T2D. Given the low r_g between glucose phenotypes and CAD, we were not surprised to observe such low predictive power.

Unless the referee and/or Editor insist, we believe that these additional analyses are beyond the scope of our study. Therefore, we have not integrated them into our revised manuscript.

	FG	mega-RG	meta-RG	meta-analysis of glucose
Genetic correlation with CAD				

	0.1476 (0.0317)	0.1043 (0.0269)	0.0788 (0.0256)	0.1060 (0.0243)
Prediction of cardiovascular disease risk using glucose PGS in GERA (AUC)				
Predictor based on COJO (s.e.)	0.5012 (0.0029)	0.5035 (0.0029)	0.5028 (0.0029)	0.5034 (0.0029)
Predictor based on SBayesR (s.e.)	0.5101 (0.0029)	0.5048 (0.0029)	0.5043 (0.0029)	0.5087 (0.0029)

4. I would be curious to see if using a PGS with only those SNPs that provide consistent direction of effects on T2D risk (increasing glucose levels allele increase risk for T2D, without considering significance of course), would have an improved predictive power.

We refer the reviewer to our answer to their Question 2 above. Overall, we observed a slight increase in the predictive power.

5. The authors are invited to explicit how they came with the estimation about the likelihood that a third of the future FG-associated SNPs to be discovered within 1Mb of the currently described loci obtained from this meta-analysis.

We apologise for the lack of clarity of this sentence. This conclusion is based on our estimates of partitioned SNP-based heritability (**Figure 4**), which shows that ~10% of FG variance is explained by SNPs within 1 Mb of the 156 COJO SNPs identified in this study. Given that these COJO SNPs only explain ~3% of FG variance, we can expect that other associations, accounting for the difference between the 10% expected and the 3% already explained, remain to be detected within 1 Mb of these 156 COJO SNPs. Next, we compared the 7% (=10%-3%) missing heritability contained within 1 Mb of COJO SNPs with the genome-wide SNP-based heritability of ~20% (in Lifelines) to get a final estimate of 7/20, i.e., ~1/3rd. The latter estimate is based on the additional (implicit) assumption that SNP-traits associations are randomly distributed across genome. However, given that this assumption has been challenged for many complex traits (e.g., heritability is concentrated in evolutionary conserved regions⁸), we have decided to remove the “1/3rd prediction” while still emphasizing that additional associations remain to be detected with 1 Mb of these 156 COJO SNPs (Lines 298–301): “Nevertheless, this analysis suggests that additional associations, accounting for the difference between the ~10% of FG variance expected and the ~3% already explained, remain to be discovered within 1 Mb of the 156 COJO SNPs identified in this study.”

6. Could the authors confirm there are no existing eQTL datasets for a more relevant tissue to glucose homeostasis other than blood? (e.g pancreatic islets, muscle, adipose tissue, where one putative gene (*CREB3L4*) seem to be mentioned as likely to regulate adipogenesis). The functional annotation part could gain in interest by integrating more data.

We thank the referee for this suggestion. Accordingly, we extended our Summary-data based Mendelian Randomization (SMR) analyses, which now includes eQTL data from 3 studies based on multiple tissues. These additional analyses have led to identify 185 unique genes showing evidence of a transcriptomic relationship with glucose levels. We further used these 185 genes to prioritize pathways and report our findings in the following (revised) section (Lines 218–284):

“Prioritisation of glucose-related genes and pathways

We performed a summary data-based mendelian randomization (SMR) analysis⁹ to prioritise genes, which mRNA expression could mediate associations between SNPs and glucose. For these analyses, we used multi-tissue expression quantitative trait loci (eQTLs) identified in the eQTLGen study¹⁰ ($N=31,684$ whole blood samples), the GTEx study¹¹ ($N=838$, across 49 different tissues) and the InsPIRE study¹² ($N = 420$ pancreatic islets samples). Using GWAS data from our largest meta-analysis of glucose ($N = 519K$), we prioritised 185 genes passing both the SMR and HEterogeneity In Dependent Instruments (HEIDI) tests ($P_{SMR} < 3.20 \times 10^{-6}=0.05/15,645$ and $P_{HEIDI} > 0.01$; **Methods**), suggesting increased evidence of a pleiotropic or a causal effect of these genes on glucose levels. A complete list of these 185 genes (hereafter referred to as SMR genes) is provided in **Supplementary Table 11**. This list includes *GCK*, *NFX1*, *CGREF1*, *CCNE2*, *QPCTL*, *ABHD1*, *SLC39A13*, *SLC12A4*, *YWHAB*, *ACVR1C*, *TRIM59*, *ITFG3*, *SMC4*, *INTS8*, *TP53INP1*, *ZCWPW1*, *KLHL42* and *SYNM* previously associated with glucose measurements, insulin measurements and T2D. Another number of SMR genes had no prior evidence of any role in glucose metabolism, but have been implicated in glucose regulation. Those include *HBM*, *CREB3L4*, *NPEPPS*, *HEXIM2*, *LCAT*, *UNC13D*, *CHMP4B*, *MTMR3*, *RCCD1*, as well as several long non-coding RNAs and pseudogenes (**Supplementary Table 11**). Among those genes, *CREB3L4* was reported to negatively regulate adipogenesis when expressed in adipose tissues¹³. Importantly, adipose tissues are involved in insulin resistance and T2D through adipokines secretion affecting systemic glucose homeostasis¹⁴. Therefore, differential expression of *CREB3L4* may change adipokine profile and, thereby, alter insulin sensitivity (**Supplementary Figure 4**).

Next, we compared our SMR results across tissues and found that 108/185 (i.e., ~58%) SMR genes had a significant effect size in at least two tissues. The remaining 77 SMR genes were more often associated with expression in pancreatic islets (16 genes), blood (15 genes) and testis (10 genes), although this enrichment was not statistically significant (Fisher Exact Test $P > 0.7$). We then focused on the 108 SMR genes detected in at least two tissues and quantified the heterogeneity of estimated effect size of gene expression on glucose levels. Overall, we found consistent effect sizes of gene expression across tissues (median Cochran's heterogeneity $I^2 \sim 40\%$). However, we also observed 12/108 genes (*KLHL42*, *STEAP1*, *MBTPS1*, *TAPBP*, *TP53INP1*, *SMC4*, *TMEM45A*, *PLEKHM1*, *CCNE2*, *ZCWPW1*, *PABPC1L*, *YWHAB*), which estimated effects had inconsistent direction across tissues. For instance, *TMEM45A* expression in pancreas, pancreatic islets, pituitary, spleen and blood was positively associated with glucose, while expression in omentum, artery (aorta and tibial), spinal cord and cultured fibroblast was negatively associated with glucose. This pattern can be explained by various mechanisms causing differential regulation of gene expression across tissues including the fact that eQTLs can have opposite effects across tissues as reported previously¹¹. For example, the G allele at *rs4132537*, an eQTL for *TMEM45A* expression, has opposite effects on gene expression in arteries and pituitary. Besides these 12 genes displaying heterogeneous effects on glucose, *GCK* showed the largest coefficient of variation of effect sizes across 7 tissues although estimates were consistently positive (**Supplementary Table 11**). Importantly, *GCK* also had the largest effect size on glucose levels ($b_{SMR} = 0.27$ mmol/L per SD of *GCK* expression in blood; S.E.=0.03; $P_{SMR}=1.6 \times 10^{-19}$, $P_{HEIDI}=0.012$), consistent with its glucose sensing role¹⁵.

Finally, we used the GENE2FUNC module of the online FUMA GWAS platform¹⁶ to annotate SMR genes with biological and functional information. Overall, we found that SMR genes are down-regulated in liver, muscle, pancreas, heart and kidney (**Supplementary Figure 5**), and significantly (False discovery rate < 5%) enriched among genes involved in peptidase activity (**Supplementary Figure 6A**), vacuole and endoplasmic reticulum membrane organisation (**Supplementary Figure 6B and 6C**). Moreover, SMR genes were significantly enriched among genes involved in myogenesis, *MTORC1* signalling, as well as within pathways related to myometrial relaxation/contraction and G-protein-coupled receptors (in particular class B secretin-like family) activity (**Supplementary Figure 7**). We also compared biological and functional enrichments of SMR genes identified through analyses of FG ($N=23$ genes; **Supplementary Table 11**) and our meta-RG ($N=148$ genes; **Supplementary Table 11**) GWAS, but did not find a significant differential enrichment between these two sets of genes.

In summary, we prioritise here 185 genes which expression across multiple tissues may explain how SNPs can induce physiological glucose variation. Further functional experiments are required to fully characterise a potential causal relationship between steady state gene expression of these genes and glucose levels.”

7. Please comment on HbA1c levels availability in UKBB and if yes, consider and discuss on the interest in performing a genetic comparison with RG in the same way they compared to FG.

We thank the referee for this question although we are not sure of what is suggested here. HbA1c measures are available in the UKB. We analysed this phenotype to detect heritability differences between fasting time groups (as we did for RG). We did not find any significant difference, implying no expected gain in using a meta-GWAS over a mega-GWAS. This observation (previously reported in **Supplementary Table 12**) is expected as HbA1c reflects glucose variation over a longer period of time (past 2-3 months).

We estimated the genetic correlation between HbA1c and FG to be ~ 0.52 (0.07), which suggests significant genetic overlap between the two traits but much less than between FG and RG ($r_g \sim 0.8$). Therefore, directly meta-analysing FG and HbA1c would not systematically improve statistical power to detect associations for each of these two traits. Other methods such as MTAG can improve power of GWAS by combining multiple genetically correlated traits. We now mention this in the discussion (Lines 311–313): “GWAS power to detect associations with FG can be further improved using statistical methods integrating GWAS data from genetically correlated traits like T2D or HbA1c¹⁷.”

Reviewer #2 (Remarks to the Author):

J. Qiao and colleagues performed an exciting study on the genetic architecture of fasting and random glucose.

We thank Reviewer #2 for their thorough revision of our manuscript and for making specific suggestions, each of which is addressed below. We also thank them for expressing excitement towards our study.

1-The authors may precise in the introduction that the genetic architecture of fasting glucose level and type 2 diabetes is only partially overlapping (Bouatia-Naji et al., Science 2008, Bouatia-Naji et al., Nat Genet 2009).

We thank the referee for underlining this. We now mention the partial overlap with type 2 diabetes in our introduction (Lines 51–57) supported by the two suggested references. “In particular, genome-wide association studies (GWAS) of glycemic traits have provided insights into the genetic regulation of glucose levels and that of T2D susceptibility^{2,18,19}, while revealing a partial overlap between them^{20,21}. Overall, GWAS of glycemic traits have discovered a range of genetic loci associated with fasting glucose (FG) concentration^{18,19,22-25}, post-challenge glucose concentration² and fasting insulin concentration^{19,23,25}”

2-I am surprised by the huge number of related individuals excluded from the analysis (N=86,465) in the UK Biobank. Many GWAS have been previously performed in the UK Biobank, and I never saw that many individuals excluded from the analysis because of relatedness issues. The authors may consider using a more relaxed threshold of relatedness (e.g. exclusion of 1st and 2nd degree relatives) to improve statistical power. Another option can be to adjust the GWAS analysis for relatedness, as previously described in literature.

The number of relatives excluded in our analysis is consistent with other UKB-based studies. For example, Loh et al. (2018) introduced BOLT-LMM in their paper reporting 459K participants with European ancestries including a subset of 337K unrelated White British samples (i.e., > 100K samples excluded)²⁶. Our definition of relatedness is based on a cut-off of 0.05 on the estimated SNP relatedness, which is slightly more stringent than 0.0625 corresponding to 2nd degree relatives. We used this definition in previously published studies (e.g., Jiang et al. *Nature Genetics*. 2019²⁷, Truong et al. *Nat Commun*. 2020²⁸, Hivert et al. *AJHG*. 2021²⁹, and Wainschtein et al. *Nature Genetics*. 2022³⁰) as this is the recommended threshold for estimates of SNP-based heritability not be confounded with shared environmental effects. Estimation of heritability (across fasting time) is central in our study design hence the use of this threshold. We agree with the referee that there are methods in the GWAS literature (e.g., BOLT-LMM) to account for relatedness. However, given our study design, we still would have needed to exclude related individuals across fasting time groups to ensure no inflation of false positives in our meta-GWAS analysis.

In summary, we acknowledge that our criterion to exclude relatives is conservative (although not unconventional), yet led to demonstrate an increase of power of a meta-GWAS over a mega-GWAS approach.

3-Related to my previous comment, did the authors apply similar exclusion procedure for related individuals in the MAGIC sample.

The MAGIC consortium consists of a large number of participating cohorts and each participating cohort has been applied with study-level quality control procedures. Therefore, there is no single criterion used. For example, a 2021 MAGIC study reports that the first- OR second-degree relatives in each cohort were excluded during their sample QC checks unless by design¹. Another 2021 MAGIC reports that related samples were excluded separately in each dataset²², but no specific thresholds were given neither in the main text nor in their Supplementary Data 1. As detailed above, we applied a relatedness threshold slightly stricter than 2nd degree relatives to minimize confounding of our estimates of SNP-based heritability.

4-The authors defined five groups based-on their self-reported fasting time (0-2h, 3h, 4h, 5h and 6-24h) to ensure that at least 30K participants were included in each group. They may also consider physiological

considerations (e.g. shape of post-prandial glucose curves) to reach an optimal definition of groups. Glucose values change a lot during the 2h postprandial period (in relation to the 1st and 2nd phase of insulin secretion), and it seems a little bit counterintuitive to have a 0-2h group.

We thank the referee for this comment. We acknowledge that substantial changes in glucose levels (and likely regulation) occur within 2h post meal. In fact, we previously highlighted this as a limitation in our Discussion: “Secondly, by choosing to analyse groups with at least 30,000 individuals, participants within 2h-post meal were aggregated in a single group, and therefore we could not assess differences in SNP effects within this critical time window.” We initially defined these groups to ensure sufficiently large sample sizes ($N > 30K$) to detect heritability and genetic correlation differences. We therefore aggregated the 0-1h group ($N=13,325$) with the 2h group ($N=59,494$) for that reason, noting that the 0-1h group represents less than 20% of the 0-2h group. Following the referee’s comment, we assessed the genetic correlation within the 0-2h group by comparing 0-1h ($N=13,325$) vs 2h ($N=59,494$). We used GCTA instead of LD score regression to estimate this genetic correlation because the former yields more precise estimates with smaller sample sizes³¹. Although, standard errors remain large, we found a genetic correlation $r_g=0.60$ (S.E. 0.17; 95%CI: 0.26-0.93) significantly lower than 1 ($P=0.018$), suggesting some heterogeneity of SNP effects between these two groups. However, there was no significant heritability difference between these groups (0-1h: $h^2=0.072$, S.E. 0.026; 2h: $h^2=0.066$, S.E. 0.007) implying that splitting the 0-2h group in two may not further improve statistical power.

We have expanded our discussion of this limitation by including the results above (Lines 344–359):

“Secondly, by choosing to analyse groups with at least 30,000 individuals, participants within 2h-post meal were aggregated in a single group despite the substantial changes in glucose levels occurring in that critical window (**Supplementary Figure 8**). Although the vast majority (~82%) of participants in that group reported their last meal exactly 2h prior to the assessment, we sought to quantify the genetic correlation of glucose levels between the 0-1h ($N=13,325$) and the 2h group ($N=59,494$). We used GCTA instead of LDSC to estimate this genetic correlation because the former yields more precise estimates with smaller sample sizes³¹. While standard errors remain large, we found a genetic correlation $r_g=0.60$ (S.E. 0.17; 95%CI: 0.26-0.93) significantly lower than 1 ($P=0.018$), suggesting some heterogeneity of SNP effects between these two sub-groups. However, there was no significant heritability difference between these groups (0-1h: $h^2=0.072$, S.E. 0.026; 2h: $h^2=0.066$, S.E. 0.007) implying that splitting the 0-2h group in two sub-groups may not further improve statistical power of our meta-GWAS. Nevertheless, if sample size is large enough, more loci showing heterogeneity of SNP effects on glucose levels can be detected within that time interval.”

5-The authors may precise the ancestry of the Lagou et al. sample (UK Biobank is predominantly EU).

We have now clarified this in the manuscript at lines 110-111: “Lagou et al. (2021) ($N = 151,188$ European ancestry individuals without diabetes).”

6-For clarity, please define in the text what means LDSC intercept.

We have now clarified this in the main text (Lines 129–135): “The mean chi-square association statistic was 1.46 in the mega-GWAS vs. 1.53 in the meta-GWAS. We also compared the LDSC intercept (a statistic reflecting the degree of confounding in a GWAS; **Methods**) between these analyses and found that both analyses yielded similar estimates of the LDSC intercept (**Table 2, Supplementary Table 5**), suggesting that the increased chi-square statistic observed in our meta-GWAS reflects enhanced statistical power but no inflation of false positives.”

And in the method section (Lines 451–453): “The LDSC intercept approximates the mean χ^2 association statistic at SNPs not associated with the trait, and therefore provides a quantification of confounding due to population stratification³².”

7-‘Overall, we found no significant evidence of heterogeneous SNP effects across fasting time groups except for rs1881415 ($P_{HET} < 5 \times 10^{-8}$), which only showed a significant effect on glucose levels within 3h post-meal (Supplementary Table 7). Importantly, this intergenic variant is in highly LD ($r^2 > 0.8$) with SNPs previously associated with fasting proinsulin and T2D risk’. The SNP *rs1881415* is also in strong LD with another SNP strongly associated with fasting glucose in the European population (Chen et al., Nat Genet 2021). This is somewhat discordant with the author’s findings and should be mentioned and discussed.

We thank the referee for this question, which also echoes another comment made by the first referee. We refer this reviewer to our response above to Reviewer #1’s first comment. In brief, our observation is consistent with what was previously reported, i.e., that the T2D risk allele at this locus is both associated with an increased FG and a decreased 2h-glucose. In our analyses, we had a limited number of fasting participants (e.g., > 5h fasting time), which limits our power to replicate the association with FG. However, we find a sign consistent pattern (**Supplementary Table 7**). We have rephrased this paragraph entirely (lines 140 – 147): “We found genome-wide significant evidence of heterogeneous SNP effects across fasting time groups for rs1881415 ($P_{HET} < 5 \times 10^{-8}$; **Supplementary Table 7**). This variant is in high LD ($r^2 > 0.8$) with SNPs previously associated with fasting glucose, fasting proinsulin, 2h-glucose¹ and T2D risk³. Importantly, the T2D risk allele at this locus shows opposite effects on FG (positive effect) vs 2h glucose levels (negative effect). Our data recapitulates this heterogeneous pattern (i.e., effect sizes are not sign-consistent across fasting time) although the relatively small number of UKB participants fasting for > 5 hours reduces the statistical power to detect a genome-wide significant effect.”

8-‘Note, as a benchmark, that this estimate is similar to the 170 LDSC genetic correlation between males and females for traits like body mass index (BMI) and waist circumference’. I think this sentence may be out of context. Please consider removing.

We thank the referee for this comment. We added this sentence to underline that meta-analysing traits with a genetic correlation of ~0.8 is not unusual as this is what we do when combining data from males and females. We apologise if this was confusing. We have now removed the sentence.

9-‘We also quantified the ability of our FG and RG PGSs to predict T2D in an independent sample of 6,905 cases and 46,983 controls from the Genetic Epidemiology Research on Adult Health and Aging (GERA) study cohort’. As GERA is multiethnic, the authors may consider performing the T2D genetic prediction study on the EU subset of GERA participants for consistency, as they identified FG and RG-associated SNPs in populations of EU ancestry.

Our analyses in GERA were already restricted to European ancestries participants. We have re-emphasized this the **Methods** section (Lines 556–558): “Our analysis is focused on individuals of European ancestry and excluded related individuals at a genetic relatedness threshold of 0.05. After QC, 6,905 T2D cases and 46,983 controls were retained.”

10-The authors may consider performing molecular pathway / gene function / tissue expression GWAS gene-enrichment analyses to investigate the biological mechanisms underlying the genetic architecture of FG and RG. As the genetic overlap between FG and RG is partial, they may identify different mechanisms involved in FG or RG genetic variation.

We thank the reviewer for this suggestion. We have now performed a pathway and gene function enrichment analysis based on genes prioritised using Mendelian Randomization. We update the method section (Lines 510–513 and 518–528) and report these additional results in the main text (Lines 273–286) and above in response to point #6 from Reviewer 1. However, given the limited number of genes prioritised from FG GWAS (Lagou et al. (2021), $N=23$), we lacked power to detect a significant differential enrichment as compared to the $N=148$ genes prioritised from our meta-RG GWAS.

11-The authors did not perform post-GWAS in vitro / in vivo studies (e.g. cellular / animal studies) to investigate the role of newly discovered GWAS variants and genes in glucose homeostasis. I understand that this may be considered as beyond the scope of the current study, but they may at least mention this limitation in the discussion.

We thank the reviewer for underlining this limitation, which we now echo in our discussion section (Lines 361–363): “Lastly, we did not perform any post-GWAS in vitro or in vivo studies, which can provide valuable evidence to investigate the role of newly discovered GWAS variants and genes in glucose homeostasis.”

References

1. Chen, J. *et al.* The trans-ancestral genomic architecture of glycaemic traits. *Nature genetics* **53**, 840-860 (2021).
2. Saxena, R. *et al.* Genetic variation in GIPR influences the glucose and insulin responses to an oral glucose challenge. *Nature genetics* **42**, 142-148 (2010).
3. Mahajan, A. *et al.* Multi-ancestry genetic study of type 2 diabetes highlights the power of diverse populations for discovery and translation. *Nature Genetics* **54**, 560-572 (2022).
4. Mahajan, A. *et al.* Fine-mapping type 2 diabetes loci to single-variant resolution using high-density imputation and islet-specific epigenome maps. *Nature genetics* **50**, 1505-1513 (2018).
5. Kycia, I. *et al.* A common type 2 diabetes risk variant potentiates activity of an evolutionarily conserved islet stretch enhancer and increases C2CD4A and C2CD4B expression. *The American Journal of Human Genetics* **102**, 620-635 (2018).
6. Xue, A. *et al.* Genome-wide association analyses identify 143 risk variants and putative regulatory mechanisms for type 2 diabetes. *Nature communications* **9**, 1-14 (2018).
7. Wang, Y. *et al.* Theoretical and empirical quantification of the accuracy of polygenic scores in ancestry divergent populations. *Nature communications* **11**, 1-9 (2020).
8. Finucane, H.K. *et al.* Partitioning heritability by functional annotation using genome-wide association summary statistics. *Nature genetics* **47**, 1228-1235 (2015).
9. Zhu, Z. *et al.* Integration of summary data from GWAS and eQTL studies predicts complex trait gene targets. *Nat Genet* **48**, 481-7 (2016).
10. Vosa, U. *et al.* Large-scale cis- and trans-eQTL analyses identify thousands of genetic loci and polygenic scores that regulate blood gene expression. *Nature Genetics* **53**, 1300-+ (2021).
11. Consortium, G. The GTEx Consortium atlas of genetic regulatory effects across human tissues. *Science* **369**, 1318-1330 (2020).
12. Viñuela, A. *et al.* Genetic variant effects on gene expression in human pancreatic islets and their implications for T2D. *Nature communications* **11**, 1-14 (2020).
13. Kim, T.H. *et al.* Identification of Creb3l4 as an essential negative regulator of adipogenesis. *Cell Death & Disease* **5**(2014).
14. Ahn, Y.H. A Journey to Understand Glucose Homeostasis: Starting from Rat Glucose Transporter Type 2 Promoter Cloning to Hyperglycemia. *Diabetes & Metabolism Journal* **42**, 465-471 (2018).
15. Matschinsky, F.M. & Wilson, D.F. The central role of glucokinase in glucose homeostasis: a perspective 50 years after demonstrating the presence of the enzyme in islets of Langerhans. *Frontiers in physiology* **10**, 148 (2019).
16. Watanabe, K., Taskesen, E., Van Bochoven, A. & Posthuma, D. Functional mapping and annotation of genetic associations with FUMA. *Nature communications* **8**, 1-11 (2017).
17. Turley, P. *et al.* Multi-trait analysis of genome-wide association summary statistics using MTAG. *Nature genetics* **50**, 229-237 (2018).
18. Dupuis, J. *et al.* New genetic loci implicated in fasting glucose homeostasis and their impact on type 2 diabetes risk. *Nat Genet* **42**, 105-16 (2010).
19. Manning, A.K. *et al.* A genome-wide approach accounting for body mass index identifies genetic variants influencing fasting glycaemic traits and insulin resistance. *Nat Genet* **44**, 659-69 (2012).
20. Bouatia-Naji, N. *et al.* A polymorphism within the G6PC2 gene is associated with fasting plasma glucose levels. *Science* **320**, 1085-1088 (2008).
21. Bouatia-Naji, N. *et al.* A variant near MTNR1B is associated with increased fasting plasma glucose levels and type 2 diabetes risk. *Nature genetics* **41**, 89-94 (2009).
22. Lagou, V. *et al.* Sex-dimorphic genetic effects and novel loci for fasting glucose and insulin variability. *Nat Commun* **12**, 24 (2021).
23. Scott, R.A. *et al.* Large-scale association analyses identify new loci influencing glycaemic traits and provide insight into the underlying biological pathways. *Nat Genet* **44**, 991-1005 (2012).
24. Horikoshi, M. *et al.* Discovery and Fine-Mapping of Glycaemic and Obesity-Related Trait Loci Using High-Density Imputation. *Plos Genetics* **11**(2015).
25. Marullo, L., Moustafa, J.S.E. & Prokopenko, I. Insights into the Genetic Susceptibility to Type 2 Diabetes from Genome-Wide Association Studies of Glycaemic Traits. *Current Diabetes Reports* **14**(2014).
26. Loh, P.-R., Kichaev, G., Gazal, S., Schoech, A.P. & Price, A.L. Mixed-model association for biobank-scale datasets. *Nature genetics* **50**, 906-908 (2018).
27. Jiang, L. *et al.* A resource-efficient tool for mixed model association analysis of large-scale data. *Nature genetics* **51**, 1749-1755 (2019).
28. Truong, B. *et al.* Efficient polygenic risk scores for biobank scale data by exploiting phenotypes from inferred relatives. *Nature communications* **11**, 1-11 (2020).

29. Hivert, V. *et al.* Estimation of non-additive genetic variance in human complex traits from a large sample of unrelated individuals. *The American Journal of Human Genetics* **108**, 786-798 (2021).
30. Wainschein, P. *et al.* Assessing the contribution of rare variants to complex trait heritability from whole-genome sequence data. *Nature Genetics* **54**, 263-273 (2022).
31. Bulik-Sullivan, B. *et al.* An atlas of genetic correlations across human diseases and traits. *Nature genetics* **47**, 1236-1241 (2015).
32. Bulik-Sullivan, B.K. *et al.* LD Score regression distinguishes confounding from polygenicity in genome-wide association studies. *Nat Genet* **47**, 291-5 (2015).

REVIEWERS' COMMENTS

Reviewer #1 (Remarks to the Author):

I thank the authors for addressing all of my main comments. I specifically appreciate the annotation provided now which I find very informative and complete.

I have no further comment, and leave the call to the editor regarding the relevance to include the predictive power estimation of RG and FG for cardiovascular disease.

Reviewer #2 (Remarks to the Author):

The authors addressed carefully all my comments, thank you. They may just consider defining the abbreviation LDSC in the text for clarity (linkage disequilibrium score regression (LDSR or LDSC)).

REVIEWERS' COMMENTS

We thank the two referees for their valuable suggestions to improve our manuscript.

Reviewer #1 (Remarks to the Author):

I thank the authors for addressing all of my main comments. I specifically appreciate the annotation provided now which I find very informative and complete. I have no further comment, and leave the call to the editor regarding the relevance to include the predictive power estimation of RG and FG for cardiovascular disease.

We thank Reviewer #1 for acknowledging the completeness of our response. We believe that these additional results could be accessed in our review (if published), as they remain outside the score of our study.

Reviewer #2 (Remarks to the Author):

The authors addressed carefully all my comments, thank you. They may just consider defining the abbreviation LDSC in the text for clarity (linkage disequilibrium score regression (LDSR or LDSC)).

We thank Reviewer #2 for appreciating our revised manuscript. We have opted for the use of "LDSC" to refer to the LD score regression method as that is also the name of the software implementing it. We have updated the manuscript accordingly.